# Point-Prevalence Survey of Antimicrobial Use in Benin Hospitals: The Need for Antimicrobial Stewardship Programs

**DOI:** 10.3390/antibiotics14060618

**Published:** 2025-06-18

**Authors:** Sarah Delfosse, Carine Laurence Yehouenou, Angèle Dohou, Dessièdé Ariane Fiogbe, Olivia Dalleur

**Affiliations:** 1Faculté de Pharmacie et des Sciences Biomédicales (FASB), Université Catholique de Louvain (UCLouvain), 1348 Brussels, Belgium; olivia.dalleur@uclouvain.be; 2Pharmacy, Clinique Saint-Luc Bouge (SLBO), 5004 Namur, Belgium; 3Clinical Pharmacy and Pharmacoepidemiology Research Group (CLIP), Louvain Drug Research Institute (LDRI), Université Catholique de Louvain (UCLouvain), 1348 Brussels, Belgium; carine.yehouenou@uclouvain.be (C.L.Y.); angele.dohou@uclouvain.be (A.D.); dessiede.fiogbe@uclouvain.be (D.A.F.); 4Laboratoire Supranational de Reference des Mycobactéries (LRM), Cotonou BP 817, Benin; 5Pharmacy, Cliniques Universitaires Saint-Luc, Université Catholique de Louvain (UCLouvain), 1348 Brussels, Belgium

**Keywords:** point prevalence survey, prescription, antibiotic resistance, AWaRe classification, Benin

## Abstract

**Background:** Antimicrobial resistance (AMR) is a public health concern worldwide, particularly in low-to-middle-income countries with few antimicrobial stewardship programs and few laboratories equipped for diagnosis. **Methods:** As point-prevalence surveys (PPSs) are a well-known tool for assessing antimicrobial use, we adjusted standardized Global-PPS for use in two hospitals in Benin and included an analysis based on the 2021 WHO AWaRe classification. **Results:** Of the 450 patients enrolled, 148 received antimicrobials (AMs) (overall prevalence 32.9%), most of them orally (54.2%). Both hospitals had a high rate of Access and Watch antibiotics use, and both prescribed mainly metronidazole. In four prescriptions, hospital A used a non-recommended association of antibiotics, such as ceftriaxone + sulbactam and ofloxacin + ornidazole. While hospital A prescribed predominantly amoxicillin + clavulanic acid (19/92; 21%) and ceftriaxone (14/92; 15%), hospital B prescribed ampicillin (24/120; 20%) and cefuroxime (14/120; n = 12%). In hospital B, surgical antimicrobial prophylaxis (SAP) was suboptimal. While there were no single-dose prophylaxis prescriptions, all one-day prophylaxis (SP2) involved ampicillin for cesarean sections. In patients in intensive care units, prolonged prophylaxis (>1 day, SP3) accounted for all postoperative prescriptions. **Conclusions:** These findings highlight the critical need for implementing antimicrobial stewardship programs, expanding diagnostic laboratory capacity to minimize empirical prescribing, and strengthening medical student training to ensure quality and rational antibiotic use, thereby addressing the growing challenge of resistance in resource-limited settings.

## 1. Introduction

Antimicrobial resistance (AMR) is a major public health concern worldwide, particularly in developing countries in which there are no antimicrobial stewardship programs (ASPs) and only poor access to last-resort antibiotics [1,2]. AMR complicates patient management and leads to higher morbidity and mortality rates [3,4]. Its rise—primarily driven by the misuse and overuse of antimicrobials in hospitals—is further aggravated by insufficient monitoring of antimicrobial use and resistance [5,6,7]

In Benin, a national action plan for tackling this situation was developed between 2019 and 2024 along the lines of the five strategic objectives of the global action plan proposed by the World Health Organization (WHO) and other international organizations in 2015. However, so far, very few activities of AMR surveillance [1,8,9] have been carried out. As in the rest of Africa [10,11], there has also been very little success with multidisciplinary antimicrobial stewardship programs (ASPs) [12], which aim to gather interventions for improving the use of antimicrobials and for better-matching antibiotics to patients [13].

Although numerous studies in developing countries have documented antimicrobial use in hospital settings, data from Benin remain scarce. This is despite the implementation of a multidisciplinary strategy for infection prevention and control: the MUSTPIC project. Targeting six hospitals and two departments (surgery and obstetrics) between 2018 and 2022, MUSTPIC aimed to promote rational antibiotic use through interventions such as hand hygiene audits [1], adherence to antimicrobial prophylaxis guidelines [10,11], mapping of resistance genes [12,13,14,15,16], and the implementation of a national guideline for the management of surgical site infections. Despite these efforts, local studies have demonstrated that antimicrobial prescribing practices in Benin remain suboptimal [14]. For instance, a recent investigation into antibiotic prophylaxis practices in gastrointestinal surgery across five hospitals in southern Benin revealed significant non-compliance with existing guidelines [15]. Alarming rates of antimicrobial resistance (AMR) have also been reported, particularly concerning commonly used antibiotics.

As structural barriers such as limited diagnostic capacity, lack of universal health insurance, and inadequate training on rational antibiotic use have been shown to contribute to the widespread use of empirical treatment in Benin [16], there is an urgent need to strengthen antimicrobial stewardship programs, improve diagnostic infrastructure, and provide comprehensive training for healthcare professionals to combat AMR.

In view of the poor documentation on antibiotic prescribing practices and antimicrobial stewardship (AMS) activities in Benin, we found it important to organize a point-prevalence survey in two public hospitals. As well as evaluating the prevalence of antimicrobial use, this was intended to identify the antimicrobial agents commonly prescribed, their prescribing patterns, and the indications for the antimicrobial prescriptions. PPS surveys are recommended by the WHO as they encourage standardization and improve the comparability of antimicrobial use over time and between hospitals [17]. They are also well known and widely used in developing countries and in low-to-middle-income countries with a high burden of infectious diseases and limited data on antibiotic consumption and resistance [10,11].

To further characterize antimicrobial prescribing trends, we applied the WHO AWaRe classification, which categorizes 180 antibacterials (ATC J01) into three groups—”Access,” “Watch” and “Reserve”—based on their potential for inducing and/or propagating resistance. The idea underlying this classification is that the most common infectious diseases could be treated empirically with 60% of the antibiotics in the Access group [18]. Finally, we assessed compliance with international guidelines, analyzed the costs and accessibility of antimicrobials available on the local market, and explored opportunities for improving the implementation of AMS in the country.

## 2. Materials and Methods

### 2.1. Study Setting and Design

Benin’s healthcare system is structured as a pyramid with three levels. The base consists of primary care facilities that provide initial contact with the population. The intermediate level includes referral hospitals, while the top level comprises national hospitals, which include teaching institutions offering a wide range of specialized medical and surgical services.

This study was conducted from February to March 2024 in the emergency departments of two public hospitals in Cotonou that offer various surgery services for outpatients and inpatients. The first involved in the emergency department in hospital A (Centre Hospitalier Universitaire de Zone Suru-Léré (CHUZ-SL)), a 101-bed hospital. The second is the emergency department in hospital B (Centre Hospitalier Universitaire de la Mère et de l’Enfant Lagune (CHU-MEL)), a 323-bed hospital specializing in gynecology and obstetrics. The adult emergency unit at hospital B consisted of two sub-units: “Emergency,” with two gynecological examination beds, and “Intensive Care,” with 16 beds used for stabilizing patients before and/or after hospitalization, surgery, or discharge.

In the absence of an informatic database, all data—including patient demographics, relevant clinical history (e.g., recent hospitalizations or procedures), prescribed antibiotics, relevant diagnoses, microbiological culture, and susceptibility test results—were collected manually.

This prospective observational multicenter cohort study was based on the existing Global-PPS protocol. The data included details on antimicrobial agents, reasons and indications for treatment and a set of quality indicators. We added data, including the prescriber category and AWaRe classification (https://aware.essentialmeds.org/groups, accessed on 4 May 2025) (Appendix A). The quality indicators serve to evaluate adherence to established antimicrobial prescribing guidelines. They encompass the documentation of clinical indications, the appropriateness of antimicrobial selection, dosing regimens and treatment duration. These parameters are instrumental in identifying deviations from optimal practices, and they facilitate the implementation of targeted strategies aimed at enhancing antimicrobial prescribing practices. The adapted Global-PPS protocol ensures a standardized method of analysis [18] and provides the starting point for another project on implementing antimicrobial stewardship programs in the country.

To assess the cost and accessibility of the AMs available in Beninese pharmacies, we compared the prices of various oral and intravenous AMs available on the official Beninese market by analyzing data from the websites of the central pharmaceutical registration and supply agency (Société Béninoise pour l’Approvisionnement en Produits de Santé SOBAPS).

### 2.2. Data Collection, Inclusion and Non-Inclusion Criteria

We included all the inpatients admitted to an intake room in the departments during the period from 12 February to 24, 2024, at hospital A and from 26 February to 9 March, 2024, at hospital B. If a patient who had already been included in this study was seen again in the following days or at another subunit, he or she was included again as a new patient only if new medical data relevant to the investigation was available (new prescription of an AM, change in the dosage of an AM or new clinical or laboratory data). Patients treated with a topical antimicrobial agent (cream, eye drops, patch, etc.) were considered not to be receiving systemic antimicrobial therapy. The following patients were not included: those attending the emergency department for a specialist consultation (e.g., gynecologic consultation) conducted by a physician other than the emergency physician on duty; those returning to the emergency department electively for a test or examination; those admitted to the emergency department without medical records (=lost to follow-up); or those who had died prematurely before any medical examination.

From 8.30 a.m. to 5.30 p.m. from Monday to Saturday and over a four-week period from 12 February 2024 to 9 March 2024, a trained pharmacist collected data according to the Global-PPS protocol (Appendix A). These data were recorded in a Microsoft^®^ Excel file (Version 16.83).

### 2.3. Data Analysis

Compliance with the guidelines according to the diagnosis and the prescriber’s indications was assessed in line with either the local guidelines included in the MUSTPIC guide for antibiotic prophylaxis (https://www.uclouvain.be/en/research-institutes/ldri/clip-in-detail, accessed on 4 May 2025) or with the international guidelines and reference articles for other infections. These guidelines are listed in Appendix A.

Descriptive analysis and statistics were performed using Microsoft^®^ Excel (Version 16.83). Parametric data were reported as mean and standard deviation, while non-parametric data were presented as median and interquartile range (25th–75th percentile). To allow for comparative analysis of the relative frequencies of the characteristics studied, proportions are expressed as percentages.

### 2.4. Ethical Considerations

Authorization to collect medical and laboratory data was granted by the relevant hospital authorities. As the study was considered a quality improvement project within the framework of the Global-PPS, explicit approval from Ethics Board Committees was not required. The auditing pharmacist signed a confidentiality agreement, and personal data, including patients’ surnames, first names, and dates of birth, were anonymized.

## 3. Results

### 3.1. Socio-Demographics Characteristics

Of the 450 patients included in the survey, 306 came from hospital B and 144 from hospital A (54 patients excluded) (Figure 1). At hospital A, approximately as many women (45%) as men (55%) were included. In contrast, at hospital B, a center specialized in obstetrics, only women were included. The overall median age was 32 years, with a median of 38 years at hospital A and 30 years at hospital B. At hospital A, most patients (106; 74%) returned home after their consultation; 31 (21%) were admitted to the hospital, and seven (5%) were referred to another hospital. Conversely, at hospital B, most patients were admitted to the hospital (244; 80%), returning home later (62; 20%).

### 3.2. Antimicrobial Prevalence

The prevalence of antibiotics was higher among hospitalized patients (58%; 32%) than patients who returned home (38%; 15%). The combined overall prevalence of AM prescriptions in the two hospitals was 33%.

Table 1 shows the general characteristics of patients who received antibiotics. The prevalence of patients on AB was higher at hospital A (61 patients; 42%) than at hospital B (87 patients; 28%). Of the latter, 47 were in the emergency department (54%) and 40 in intensive care (46%). At 32 years, the median age of the 148 patients with AB prescription was identical to that of the overall population. Overall, 212 AB prescriptions were written. Twice as many patients were prescribed a single AB (92; 62%) than those who were prescribed two ABs (48; 32%).

The status of prescribers differed between the two hospitals. At hospital A, 91 (99%) out of 92 prescriptions were written by one of the department’s three general practitioners and only one by an emergency medicine resident. At hospital B, 114 (95%) out of 120 prescriptions were signed by gynecology-obstetrics residents and 6 by nurses (5%).

### 3.3. Antimicrobials Use

Of the AM prescribed in the two hospitals, metronidazole accounted for 27% of prescriptions: 22 out of 92 (24%) in hospital A and 36 out of 120 (30%) in hospital B. Hospital A prescribed more amoxicillin/clavulanic acid (19/92; 21%) and ceftriaxone (14/92; 15%) than hospital B, which prescribed more ampicillin (24/120; 20%) and cefuroxime (14/120; 12%) (Figure 2). However, direct comparisons should be interpreted with caution, as hospital B specializes in gynecology and obstetrics, whereas hospital A is a general hospital with a small emergency department.

The main routes of administration in both hospitals were oral (54.2%; 115/212) and intravenous (44.3%; 94/212). Only three prescriptions were administered vaginally (1.5%).

The main infectious conditions diagnosed by clinicians differed from hospital to hospital (Appendix A). At hospital A, there were 41 prescriptions for gastrointestinal infections (45%) and 28 for skin and soft-tissue infections (30%). At hospital B, 27 AMs were prescribed for gynecological-obstetric prophylaxis (53%) and 64 for gynecological-obstetric infections (23%). Community-acquired infections represented the most common indication for the use of AM (Table 2).

### 3.4. Antimicrobials for Prophylaxis

Of the 120 AM prescribed at hospital B, 53% (64/120) were used for surgical antibiotics prophylaxis, the most common of which is for surgical prophylaxis that lasted more than one day (SP3) (62.5%; 40/64), followed by surgical prophylaxis lasting one day (SP2) (37.5%; 24/64). There was no single-dose surgical prophylaxis (SP1). Prescriptions for one-day prophylaxis (SP2) were all for ampicillin 2 g one shot, followed by 1 g every 8 h before delivery/cesarean section. Prescriptions for prophylaxis > 1 day (SP3) accounted for 100% (40) of postoperative prescriptions in 26 patients hospitalized in Intensive Care. The treatments used in these patients were mainly 38% metronidazole alone (10/26), 31% metronidazole combined with cefuroxime (8/26), and 19% metronidazole combined with amoxicillin/clavulanic acid (5/26) (Appendix A).

### 3.5. Quality Indicators for Antimicrobial Prescribing

The quality indicators for prescriptions were well documented, ranging from 96% in hospital A to 98% in hospital B. However, the following is true:The dates for discontinuation and/or re-evaluation of prescriptions were noted in only 54% to 58% of patients’ medical records (Appendix A);The national guidelines/recommendations had been met for only 36% (76/212) of the AM prescriptions. Regarding compliance with guidelines on the choice of AM, only 44% (91/205) of prescriptions complied with international guidelines: 50% (44/88) at hospital A and 40% (47/117) at hospital B. Of the non-compliant prescriptions, 61 (51%) did not require an AM; 45 (37%) were too narrow a spectrum; and the remaining 15 (12%) were too broad spectrum.Regarding the dosage of AM, only 50% (105/212) of prescriptions complied with international guidelines. This percentage was nearly equal in the two hospitals, at 48% (44/92) (hospital A) and 51% (61/120) (hospital B);The proportion of assessable prescription durations (i.e., with a documented end date/reassessment) that were compliant varied between hospitals: 84% (42/50) at hospital A and 46% (32/69) at hospital B.

### 3.6. AWaRe Classification

Of the AM prescriptions, those in ATC category J01 (antibacterials AB for systemic use) were classified in each of the WHO AWaRe categories: Access, Watch and Reserve, Unclassified, and Not Recommended. As Table 3 shows, AB in the “Access” category was the most prescribed: 72% (76/105) at hospital B and 54% (45/84) at hospital A, i.e., 121 prescriptions out of the total of 189 (64.1%). AB in the “Watch” category were the next most prescribed: 40% (34/84) at hospital A and 27% (28/105) at hospital B. No antibiotic prescription in the “Reserve” category was reported in the two hospitals audited.

The “non-recommended” combined ABs were prescribed exclusively at hospital A, with four prescriptions (4.8%) of ceftriaxone + sulbactam and 1 prescription (1.2%) of ofloxacin + ornidazole. Only one “Unclassified” antibiotic was used in the audited hospitals: Rovamycin.

### 3.7. Antimicrobials Costs

As shown in Appendix A, our price analysis of antibacterials (ABs) for systemic use (D01) units found that the following oral ABs were less expensive than their intravenous forms.

Additionally, a comparison of daily treatment costs revealed notable differences between recommended antibiotic prescriptions and those associated with poor prescribing indicators (choice of compliant AM) (Table 4).

For treatments with an overly broad spectrum, the total daily cost reached FCFA 35,351, whereas the recommended treatments cost FCFA 45,696. Conversely, for narrow-spectrum treatments, the total daily cost was FCFA 29,213 compared with FCFA 134,568 for the recommended alternatives. Furthermore, unnecessary antibiotic prescriptions resulted in a potential saving of FCFA 79,497. These discrepancies were largely driven by the high cost of gynecological-obstetric antibiotic prophylaxis in hospital B, where ampicillin 3000 mg/day (FCFA 900) was prescribed rather than lincomycin 1800 mg/day (FCFA 5 025). Similarly, in hospital A, treatments for open wounds following road accidents involved lincomycin 1500 mg/day (FCFA 324) or amoxicillin 1000 mg/day (FCFA 196) rather than the recommended amoxicillin/clavulanic acid 875 mg/125 mg three times daily (FCFA 609). For a better interpretation, it should be noted that, depending on the source, average monthly incomes in Benin range between FCFA 73.199 and 79.252 [20]; at the time of writing, the monetary value of one euro was equivalent to FCFA 655.96.

## 4. Discussion

This study, which describes a dataset that was collected using the global PPS data-collection tool, concerned the prescription of antimicrobials to 450 adult patients at two public teaching hospitals in Benin [17]. At 32.9% (n = 148/450), the prevalence of antimicrobial use was relatively lower than the prevalences found by PPS in neighboring countries such as Nigeria in 2018 (59.1%), Zambia in 2019 (57%), Ghana in 2019 (55%), and Uganda in 2019 (45%) [21,22]. This difference may have been due to the fact that the populations studied varied between countries and that all the other studies took place four years earlier than ours. There is also the fact that economic, social, and healthcare situations differ between countries. Similarly, comparing Global-PPS data collected at different time points within the same hospitals or even the same departments remain challenging due to potential variations in patient case mix, staffing, prescribing practices, and contextual factors.

Fifty-four percent of our patients received oral antibiotics, and 44% received venous antibiotics. The most common indications for antibiotic prescription were community-acquired infections (CAIs), with gastrointestinal infections as the leading cause. The high frequency of CAIs observed in Sub-Saharan Africa can be attributed, first of all, to the high burden of infectious diseases in the region, which is in itself linked to inadequate hygiene practices and inadequate infrastructures for environmental sanitation. Secondly, it can also be explained by the fact that the departments we investigated (intake room) received a high number of patients with CAIs. Authors in neighboring Nigeria, where socio-economic problems are very similar, also found that CAIs were the main type of infection [21].

Due undoubtedly to the absence of national-level guidelines, compliance with international or institutional antimicrobial guidelines was relatively low. The guidelines elaborated by the MUSTPIC project are currently awaiting official national validation and concern only surgical-site infections in gynecology and the gastrointestinal area. Benin also lacks laboratories capable of providing reliable antibiogram results that would promote the rational prescription of antibiotics [23]. As a result, doctors must constantly treat patients empirically. Although Benin has been registered on the GLASS (Global Antimicrobial Resistance and Use Surveillance System) platform since 2020, sample collection and the implementation of surveillance procedures are still in the preparatory phase [24]. Neither does the country currently have any structured training programs on the rational use of antibiotics [17,25]. For comparison, similar training initiatives in Pakistan have led to an average guideline compliance rate of approximately 30.5%–which remains substantially lower than the rates seen in Malaysia (50.4%) and Australia (67.3%) [25,26,27].

According to the WHO AWaRe classification system, the antibiotics prescribed most in our study belonged to the Access group (64%). A previous study undertaken in a surgical setting in Benin found resistance rates to third-generation cephalosporins of over 75% in *Klebsiella pneumoniae* and 68% for extended-spectrum-beta-lactamases-producing *Escherichia coli* [28]. These high rates of resistance may be explained mainly by the fact that, at the time of this study (2020), antibiotics were purchased without a prescription.

Highly relevant in this regard is a per-country study by Pauwels I and al. involving Global PPS data from 2015, 2017, and 2018 collected in 664 hospitals in 69 countries in which the authors found large differences in antibiotic prescribing in the AWaRe group [29]. While the highest Access percentage (57.7%) and the lowest Watch percentage (41.3%) were found in Oceania, the highest Access percentages at the country level were observed in Sub-Saharan countries such as Guinea (66.7%), South Africa (61.9%), and Togo (59.8%) [29]. These data confirm the percentages we found in our own study (72% in hospital A and 54% in hospital B).

An important quality indicator in our study is the prevalence of combination therapy, the main concern regarding nonconformity being the use of metronidazole 500 g twice daily, amoxicillin/clavulanic acid 1 g/200 mg (with unclear dosing frequency), and cefuroxime 500 mg twice daily. As the antibiotics in the beta lactams group are time-dependent-killing antibiotics that seek to optimize the duration of a pathogen’s exposure to an antimicrobial, the recommendation is to reach 3–4 doses per day [30].

In the hospitals in our study, surgical antimicrobial prophylaxis (SAP) was conducted for more than 1 day in 40% of cases, and the three most common regimens of antibiotics, after the shot of ampicillin, were, respectively, metronidazole, metronidazole + cefuroxime and metronidazole + amoxicillin/clavulanic acid. According to the international recommendations, SAP should be conducted according to the five international criteria—dosage, timing, choice of antibiotics, indication, and duration—and should not exceed 24 h [31,32,33,34]. In an earlier study, in four public maternity hospitals in Benin, Dohou et al. found that SAP did not follow the international guidelines and proposed national guidelines to avoid the misuse of antibiotics in surgical departments [35]. They also showed that ampicillin, which had been used for prophylaxis so far (as of 2009), was inefficient. Almost all the bacteria isolated in surgical-site infections in the hospitals they studied were resistant to the ampicillin contained in the cesarean kit [28]. To allow better management of SAP and surgical site infections, a new booklet was developed by the team of the MUSTPIC project (https://www.uclouvain.be/en/research-institutes/ldri/clip-in-detail, accessed on 4 May 2025). It now awaits validation by the national authorities before being deployed in hospitals throughout Benin.

Most of the antibiotics in our study were prescribed without bacteriological analysis, illustrating the extent to which the poor availability of clinical laboratories in the country is a significant barrier to the surveillance and monitoring of resistance recommended by the WHO [36,37,38]. While there are 150 laboratories located in the south of Benin, for example, only 25 in the whole country focus on bacteriology. There are three main reasons empirical treatment is maintained: the absence of universal health insurance for patients, the lack of automated systems and high-level equipment that would reduce turnaround times, and doctors’ incomplete knowledge of the local hospital ecology.

The MUSTPIC project (2018–2022) was the first project to describe the bacterial ecology of surgical departments across six public hospitals in Benin [31]. The absence of systematic microbiological surveillance, limited knowledge of hospital bacterial ecology, and high antibiotic prescription rates contribute to the ongoing emergence of antimicrobial resistance. Conducting other point-prevalence surveys in larger hospitals and surgical departments could improve data representativity and provide antimicrobial stewardship strategies. In practice, antibiotic use could be enhanced by initiatives such as sharing audit reports with hospitals, investigating prescribing rationales—particularly in gynecological prophylaxis—and developing practical prescribing tools such as dosing charts for standard and renally impaired patients. As current pharmacology training is limited, it is also crucial to strengthen medical education. Finally, if curricula were revised in ways that ensured more effective integration with antimicrobial stewardship, prescribing behaviors and resistance control would undergo long-term improvements.

A key strength of this study is that it is one of the first published reports on global PPS in Benin and the first to use the AWaRe classification framework specifically to analyze antibiotic consumption. It also constitutes the preliminary steps toward the implementation of an antimicrobial stewardship program in the country. The main limitation of this study concerns the methodology of the point prevalence. Firstly, as data collection was limited to a short period, it may not fully have captured variations in antimicrobial prescribing patterns over time. Even though our findings provide a valuable snapshot of prescribing practices, fluctuations in trends across different weeks or seasons cannot be excluded. Meanwhile, point-prevalence surveys remain a widely used and informative method for assessing antimicrobial use in healthcare settings. Secondly, as the work focused only on two public hospitals (located at different levels in the overall healthcare system) and only on a small number of departments, the results could not be generalized directly to other Benin hospitals. Finally, due to practical challenges encountered in the field—such as paper-based prescribing and incomplete patient records—a repeated study would greatly enhance the reliability and robustness of our conclusions.

## 5. Conclusions

WHO-recommended targets were not reached: adherence was 44% for antimicrobial selection, 50% for dosage, and 62% for treatment duration. Our study highlights the need to implement antimicrobial stewardship in hospitals in Benin. Particular attention should also be devoted to training medical students on the importance of reducing the misuse of antibiotics through proper prescribing practice. Strengthening AMS programs would help optimize antibiotic use, reduce antimicrobial resistance, and improve patient outcomes.

## Figures and Tables

**Figure 1 antibiotics-14-00618-f001:**
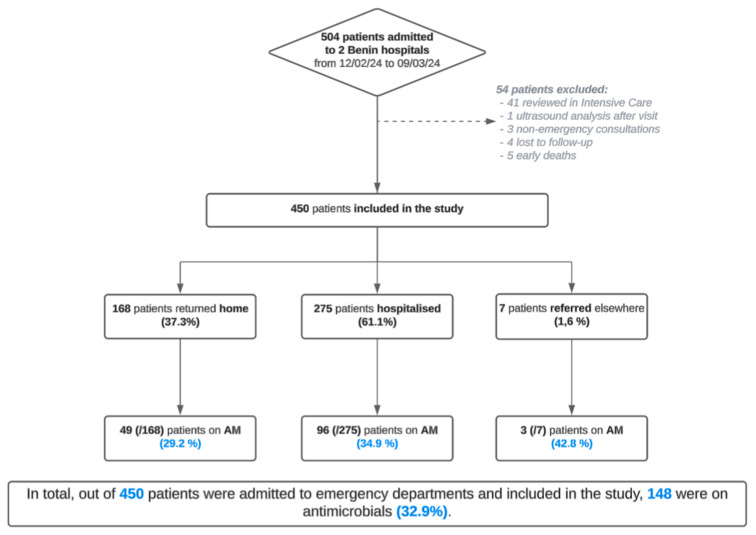
Flowchart of the various patients included in the prospective observational study in the two Beninese hospitals from 12 February to 9 March 2024 inclusive.

**Figure 2 antibiotics-14-00618-f002:**
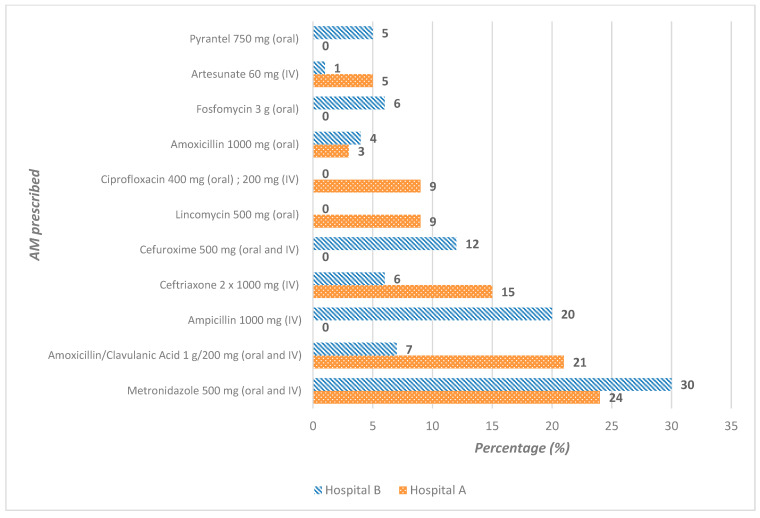
Antimicrobials commonly prescribed to patients at hospital A (N = 92) and at hospital B (N = 120), Benin.

**Table 1 antibiotics-14-00618-t001:** Specific characteristics of patients and prescriptions with antibiotics (AB).

	Population Hospital A	Population Hospital B	Total Population
	N (%) [IQ 25; IQ 75]	N (%) [IQ 25; IQ 75]	N (%) [IQ 25; IQ 75]
Total patients included	144	306	450
Patients on AB	61 (42)	87 (28)	148 (33)
- Emergency	61 (100)	47 (54)	108 (73)
- Intensive Care	-	40 (46)	40 (27)
Socio-demographic variables			
Men	31 (51)	0 (0)	31 (21)
Women	30 (49)	87 (100)	117 (79)
Median age	38 years old [27; 46]	30 years old [23; 35]	32 years old [25; 42]
Minimum age	13 years old	16 years old	13 years old
Maximum age	79 years old	43 years old	79 years old
Patient disposition			
Home	40 (65)	9 (10)	49 (33)
Hospitalization	18 (30)	78 (90)	96 (65)
Referral to another institution	3 (5)	0 (0)	3 (2)
Co-morbidities			
Patient with no co-morbidity/unknown	56 (92)	75 (86)	131 (88)
Patients with 1 co-morbidity	5 (8)	12 (14)	17 (11)
Patients with >1 co-morbidity	0 (0)	1 (1)	1 (1)
None	51 (84)	75 (86)	126 (85)
Hematological or solid cancers	0 (0)	0 (0)	0 (0)
Chronic/long COVID	0 (0)	0 (0)	0 (0)
Type I or II diabetes mellitus	2 (3)	0 (0)	2 (1)
Immunosuppressed, non-oncological	0 (0)	0 (0)	0 (0)
Unknown	4 (7)	0 (0)	4 (3)
Gastroenterological diseases	1 (2)	0 (0)	1 (1)
Lung diseases	1 (2)	2 (2)	3 (2)
Malnutrition	0 (0)	0 (0)	0 (0)
Trauma	0 (0)	0 (0)	0 (0)
HIV	1 (2)	2 (2)	3 (2)
Other	0 (0)	9 (10)	9 (6)
AM requirements			
Patients prescribed with 1 AM	35 (57)	57 (66)	92 (62)
Patients prescribed with 2 AM	21 (34)	27 (31)	48 (32)
Patients prescribed with ≥3 AM	5 (8)	3 (3)	8 (5)

AM: Antimicrobials; HIV: Human Immunodeficiency Virus.

**Table 2 antibiotics-14-00618-t002:** Divergent indications for AM prescribed at hospital A (N = 92) and B (N = 120).

	Hospital A N (%)	Hospital B N (%)	Total N (%)
Total prescribed AM	92 (43)	120 (57)	212 (100)
Therapeutics			
Community-acquired infections (CAI)	92 (100)	37 (31)	129 (61)
Healthcare-associated infections (HAI)	0 (0)	16 (13)	16 (8)
Prophylaxis			
Surgical prophylaxis (SP)	0 (0)	64 (53)	64 (30)
Single-dose surgical prophylaxis (SP1)	-	0 (0)	0 (0)
Single-day surgical prophylaxis (SP2)	-	24 (37.5)	24 (37.5)
Surgical prophylaxis > 1 day (SP3)	-	40 (62.5)	40 (62.5)
Medical prophylaxis (MP)	0 (0)	0 (0)	0 (0)
Other	0 (0)	0 (0)	0 (0)
Unknown	0 (0)	3 (3)	3 (1)

**Table 3 antibiotics-14-00618-t003:** Classification of MA ATC J01 according to AWaRe (WHO) prescribed at hospital A (N = 84) and B (N = 105).

	Hospital A N (%)	Hospital B N (%)	Total N (%)
Access	45 (54)	76 (72)	121 (64.1)
Watch	34 (40)	28 (27)	62 (32.8)
Reserve	0 (0)	0 (0)	0 (0)
Not Recommended	5 (6)	0 (0)	5 (2.6)
Not Rated	0 (0)	1 (1)	1 (0.5)

N.B.: “Not Recommended” includes fixed-dose combinations of broad-spectrum antibiotics whose use is not evidence-based [19].

**Table 4 antibiotics-14-00618-t004:** Comparative daily costs of antimicrobial prescriptions relative to compliance with guidelines.

	Non-Compliant (FCFA)	Recommended (FCFA)	Difference (FCFA)
Broad-spectrum treatments	35,351	45,696	+10,345
Narrow-spectrum treatments	29,213	134,568	+105,355
Potential savings from unnecessary AM	79,497	-	−79,497
Specific examples			
Gynecological-obstetric prophylaxis (hospital B)	900	5 025	+4125
Open wound treatment (hospital A)	324 or 196	609	+285 or +413

AM: Antimicrobials. Note: Average monthly income in Benin: FCFA 73,199–79,252.

## Data Availability

The original contributions presented in the study are included in the article. For Appendix A, please address any inquiries to the corresponding authors.

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
