# Peer review of "Point-Prevalence Survey of Antimicrobial Use in Benin Hospitals: The Need for Antimicrobial Stewardship Programs"

_antibiotics, 2025, doi:10.3390/antibiotics14060618_

Round 1

Reviewer 1 Report

Comments and Suggestions for Authors

I have re-reviewed the manuscript titled “POINT-PREVALENCE SURVEY OF ANTIMICROBIAL USE IN BENIN HOSPITALS: THE NEED FOR ANTIMICROBIAL STEWARDSHIP PROGRAMS”

The Study describes a data set, collected using the global PPS data collection tool, on the prescription of antimicrobials to 450 adult patients in the hospitalization rooms of two teaching public hospitals in Benin. AM use prevalence, use structure, prescribing quality and costs are evaluated and discussed. The evaluation of the data is very comprehensive and sheds light on various aspects. As data on antibiotic prescribing in Benin is rare, this study is relevant

However, I believe the manuscript requires some major revisions:

Introduction

the introduction is very comprehensive and detailed. It would make sense to shorten the introduction somewhat and to formulate it more specifically for the study conducted. In contrast to the interesting but very general introduction at the beginning of the paragraph, the relevant aspect, i.e. the objectives of the present study, is only mentioned in a very condensed form at the end of the introduction. The manuscript would benefit from a different emphasis in the introduction.

Methods

  • Study setting and design:

Indicators and parameters studied are listed in detail. It would be helpful to have an overview of which parameters are included in the global PPS protocol and which parameters are additionally evaluated. I would suggest providing this information in a table.

What is the evidence base for the additionally assessed parameters?

  • Line 128: please specify: “all patients”; does this include also those patients that were admitted as in-patients to the wards or only out-patients? (is depicted in figure 1, but could be mentioned within the text as well?) are children included as well? (details are given in Table 1, but could be mentioned within the text as well?)
  • Line 148/149: there is a cross link to appendix 5, but the content of appendix 5 does not correspond to the referring text. choice of substance/compliance with guidelines: please refer to the guidelines used
  • Please include information on the hospitals structure of benin within your manuscript. How many hospitals are there? Total bed capacity?... It would be helpful to understand the applicability of the presented results to other hospitals, regions, … which proportion of hospitals/hospital beds of the city/region/country are coverd by the study?

Results:

  • AM prevalence: is it always (A % ; B %)? Please specify within the description. Of the results
  • Table 1: the lines of the table are slightly shifted; please improve this to ease readability.
  • Figure 2: I would not classify the anthelmintic pyrantel as an antibiotic
  • Figure 2: rethink the design of the figure. Why different colours and different structures of the bars? I would prefer filled bars in two different colours
  • Figure 2: label of x-axis: percentage of all AM prescribed? Maybe add the caption
  • Line 203: vaginally application – systemic AM or local AM?
  • Lines 212-201: I'm not sure if I understand this correctly: 53% of AM in hospital B were prophylactic prescriptions? Or were 53% of prophylactic prescriptions in hospital B (and 47% of prophylactic prescriptions in hospital A)?. Please rephrase to enease readability
  • Section 3.5: Please restructure/rephrase the paragraph. For readers who are not so familiar with the use of QIs or AM, this paragraph would be difficult to understand at the moment. Suggestion: name the respective quality indicators (bullet points) and summarise your results for this indicator in one short sentence.

I would suggest to include a figure or a table on the QIs as this is – in my opinion - a major result of the study

It is not clear why the reference values per QI are varying (212, 205, 88, 117, … … )

Are the QIs calculated overall or separately for prophylactic and therapeutic indications? (both possible but this has to be stated clearly)

  • Table 3: please standardize decimals
  • Appendix 2: I would recommend to add the daily therapy costs
  • Section 3.7: without the respective table (appendix 2) it is hard to understand the paragraph. Please consider to add a table or a figure to this paragraph. Especially this phrase “For treatments with an overly broad spectrum, the total daily cost reached 35 351 FCFA, whereas the recommended treatments cost 45 696 FCFA. Conversely, for narrow-spectrum treatments, the total daily cost was 29 213 FCFA compared to 134 568 FCFA for the recommended alternatives. Furthermore, unnecessary antibiotic prescriptions resulted in a potential savings of 79 497 FCFA.” is not selfexplaining

Discussion:

  • Lines 272-274: were those studies performed in similar departments of the hospitals? The present study only described the prescribing behaviour in specialized departments.
  • Line 280: the high frequency of CAIs might also be influenced by the departments that were studied in this investigation. (hospitalization rooms): it seems to be a logical conclusion that a particularly large number of community-acquired infections are treated in these departments
  • Lines 301-303: what is your conclusion in connection with the statement to the resistance rates?
  • Lines 312-314: the discussion of the combination therapies should be separated from the discussion of application regimes
  • Lines 314-317: Is this part of the discussion only about the therapeutic use of the antibiotics in question or also about the SAP (where we do not have a preference for 3-4 doses per day)?
  • Please include the spezialization of the hospitals/departments in the limitation-discussion

Conclusion:

  • Depending on the proportion of hospitals or beds covered by the study (?), the conclusions might be too generalised.

References:

Line 36-38:      find better suitable (or additional) reference for this statement

Line 38-40:      find better suitable (or additional) reference for this statement

Line 82/83:     add reference for this statement

Line 112:         reference for the aware categorie: add in reference section and cite according to standards

Line 120:         reference 18 seems to be not suitable

Line 147:         add in reference section and cite according to standards

Lines 355-367: add references for your statements

Reference 27 and 42 seem to be the same

General: References should be citetd according to MDPI Reference List and Citations Style Guide. This is not the case for all cited references at the moment. Please adapt your citations to the required style. (especially 1, 4, 5, 7, 8, 10, 11, 14, 19, 22, 24, 30, 43)

Consider adding references to the appendix

Comments on the Quality of English Language

I would recommend a general language revision

Line 38:           “and” instead of “with”

Line 39:           “increases in patient care” should be changed to “to increased effort in patient care”

Line 55:           “comparability” instead of “comparisons”

Line 59:           “…only a few activities have been carried out so far…” instead of “…few activities have been conducted so far…”

Line 191:         “antimicrobial use” instead of “Antimicrobials drug use”

Line 182:         “… of the 148 patients with AB prescription” instead of “… 148 patients undergoing AB…”

General:          abbreviation and wording “antimicrobial (AM) or antibiotic (AB), please harmonize

General:          do not capitalize the word “hospital”

Line 186:         “greatly” instead of “enormously”

Line 187:         “…91 (99 %) out of 92 prescriptions were written…” instead of “…91 prescriptions (99%) out of 92 were written…”, the same applies to the following sentence

Line 191:         rephrase the title of the paragraph

Line 192:         “among” instead of “regarding”

Line 208:         community acquired infections

Line 222:         “quality indicators for antimicrobial prescribing” or something similar instead of “Antimicrobials quality indicators”

Line 240:         “and” instead of “followed by”

Line 253:         aforms

Line 278:         intraveneous

Line 303:         extended-spectrum beta-lactamase-producing

Line 305:         consider “in antibiotic prescribing” instead of  “prescription of antibiotics”

Line 310:         “confirm” (present tense)

Line 312-314:  syntax

Line 355:         “of” instead of “in”

Line 363:         “dosing chart” instead of “dosage chart”

  • Appendix 1: please translate into English

Author Response

Comments and suggestions for Reviewer 1

Thank you very much for your comments and suggestions. Please find below the detailed responses and the revisions highlighted in track changes in the resubmitted version of this manuscript. The comment about the introduction that should be shorten is taking account in the manuscript as well as the modification of certain table’s presentation.

The English also is corrected as suggested and we hope that it will be clearer to the readers. 

Study and design: it would be helpful to provide an overview of which parameters are included in the global PPS protocol, and which are additionally assessed?  What is the evidence base for the additionally assessed parameters?

Response: In our study and according to the global pps in general, we used to collect parameters for each patient on antimicrobial agents (age, sex, weight and biomarker information’s as well as bacteriological analysis) and for each antimicrobial prescription (dose, start day, route of administration, indication for treatment), reason for antimicrobial prescription and stop, adherence to local guidelines, for targeted prescriptions record the pathogens and their respective resistance type. We adapted questionnaires according to the reality and the healthcare settings

In our study we completed classification of antibiotics according to the WHO AWaRe (Access, Watch and Reserve) to propose after some recommendations to hospitals. Moreover, this audit constitutes the start point of another project: the implementation of antimicrobial stewardship programs in public hospitals. We completed also the name and the qualification of medical doctor, the diagnosis and indication of antibiotic prescription, renal function adaptation, type of microorganisms as well as their susceptibility. In the country, the antibiotics should be prescribed by all health care professional (nurse, technicians, doctor, midwives), and its important to highlight and complete this information in manuscript.

Line 128: please specify “all patients «does this include also those patients that were admitted as-in-patients to the wards or only outpatients? Are children included as well.

Response: All information will be completed in the text as suggested

Children are not included.

All patients concern the in-patients present this day and who already take antibiotics.

Line 148:  Refer to the guidelines used

Response: Thank you for this comment. The correction will be done. The content of table 5 is not the same that the idea highlights on the text. The guideline here is the MUSTPIC guideline. Previously the team have conducted project entitled multidisciplinary strategy for prevention and infection control (MUSTPIC). They described hand hygiene attitudes, antimicrobial prophylaxis challenges as well as different bacteria isolated on surgical site infections (SSIs) and they proposed national guidelines for treatment of SSIs and for prescription of antibiotic during prophylaxis.

Results:

All of comments and suggestions regarding the results will be taken account in the last version

Figure 2: I would not classify the anthelmintic pyrantel as an antibiotic.

Thank you for this comment. We want to highlight the title of this figure. We showed all the antimicrobials and not only antibiotics. We presented here the antimicrobials (antibiotics, antiparasitic such as artesunate) used per hospitals. The two types of colors represented hospital A and B. The title of the manuscript concern also antimicrobials and not only antibiotics.

Are the Qis calculated overall or separately for prophylactic and therapeutic indications?

Response: This precision will be complete on the final manuscript. The denominator depends on the hospital A or B (number of prescriptions and the number of antimicrobials used are different). A short commentary will be completed

Reviewer 2 Report

Comments and Suggestions for Authors

The authors should consider the following:

Sample size justification should be provided.

The effect size should be clearly referred to a related study, or clear explanation is needed.

Limitations of the study should be better discussed.

Missing data workflow should be clearly defined and described in the article.

Novelty of the study should be clearly stated in the abstract.

Ethnicity of the study subjects should be provided.

The authors should list clearly the possible confounding factors of the study.

IRB approval number should be provided.

The laboratory tests should be performed via a centralised laboratory in the entire study, otherwise, the authors should justify how the measurement uncertainty among interlaboratory can be established and/or adjusted.

Comments on the Quality of English Language

Improvement needed

Author Response

Comments and suggestions for Reviewer 2

Thank you very much for taking the time to review this manuscript. Please find the detailed responses below and the corresponding revisions/corrections highlighted in track changes in the resubmitted files.

We addressed the following remarks of reviewer here and completed these in the final manuscript.

  1. Samples size justification should be provided:

Response:  we selected two public and teaching hospitals in the country according to the health care system. As we decided to use the global point prevalence that constituted a validated and international tool to explore antibiotics consumption and use in hospital settings the sample size is not required. It was one day, and one point collect. We have included all the patients present this day on hospitalization room and who has already an antibiotic prescription. For this kind of study, the sample size calculation is not required.

  1. The effect size should be clearly referred to a related study or clear explanation is needed

Thank you for your valuable comment. This Global PPS audit represents the first implementation in Benin, and we indeed started with two hospitals: one focusing on general adult emergency wards, and the other on gynecology-obstetric emergency units. The decision to begin with these two different and high-demand areas was driven by the objective of obtaining a preliminary insight into antimicrobial prescription patterns in critical care settings, despite being non-extrapolatable to all hospitals in Benin at this stage. Given this exploratory context, we did not aim to calculate a specific effect size; instead, we focused on descriptive analysis of the prevalence and patterns of antimicrobial use, guided by the Global-PPS protocol. In addition, this initial study is intended to establish a baseline for future, more comprehensive national audits and for the progressive implementation of antimicrobial stewardship programs in public hospitals. In the revised manuscript, we will clearly state that the scope of the sample was limited to two hospitals for this initial phase, and thus effect size calculations were not the primary focus. Instead, we provided proportions and prevalence rates to offer a snapshot of the current state of antimicrobial use in these critical settings.

  1. Limitations of the study should be better discussed:

Response: Thank you for this comment. We will complete the limitation section in the manuscript. First its important to highlight the fact that different analysis were performed in different laboratory with the inter variability effect. Secondly, including two public hospitals cannot allow generalization of our data regarding the prescription and use of antibiotics in Benin.

  1. Missing data workflow should be clearly defined and described in the article

Response: As our knowledge, we can not identify clearly a missing data as the questionnaire used is a recommended and international proposed by the Global PPS

  1. Novelty of the study subjects should be provided:

Response: The data relative to the use of antibiotics and antimicrobial resistance is rare in Benin. Very few studies described using this recommended tool (Global pps) the prescription of antibiotics in our hospitals. Moreover, we added description regarding the WHO AWaRe classification. This study highlights the importance of implementing antimicrobial stewardship in hospitals.

  1. The authors should list clearly the possible confounding factors

Response: The possible confounding factor as our knowledge is the fact that the data were collected by one trained pharmacist, and they were no checking by another people to strengthen data.  

  1. IRB approval number should be provided:

Response: All the data in the case of Global pps was collected directly on the sheet of patients without their consent. All necessary information’s were collected about treatment and no dialogue was established between patients and professionals. However, we obtained prior to the study the agreement of hospitals as recommended by university of Antwerpen (data use agreement for noncommercial purpose between U. Antwerpen and hospitals)

  1. The laboratory tests should be performed via a centralized lab in the entire study, otherwise the authors should justify how the measurement uncertainty among interlaboratory can be established and or adjusted

Response: thank you for this comment. This point will be added in the limits of manuscript. As we worked with two different hospitals it was not possible to perform all analysis in the same laboratory. In the country, we lack national health insurance to cover all the costs related to the patient’s analysis. So, they were not forced to do complementary analysis in particular laboratory. Authors are aware about possible variability inter laboratories but the most important outcome here is the consumption and use of antibiotics. Moreover, the number of labs who performed bacteriology analysis remains scarce. Most of the antibiotic treatment remains probabilistic to the start until the end and this point is very alarming in our setting.